# Physicians’ Knowledge, Attitude and Practice of Generic Substitution in China: A Cross-Sectional Online Survey

**DOI:** 10.3390/ijerph18157749

**Published:** 2021-07-21

**Authors:** Mingyue Zhao, Lingyi Zhang, Zhitong Feng, Yu Fang

**Affiliations:** The Department of Pharmacy Administration and Clinical Pharmacy, School of Pharmacy, Xi’an Jiaotong University, Xi’an 710000, China; mingyue0204@xjtu.edu.cn (M.Z.); zly_123456@stu.xjtu.edu.cn (L.Z.); fzt5051445@stu.xjtu.edu.cn (Z.F.)

**Keywords:** knowledge, attitudes and practice, generic substitution, originator, prescribe, physician

## Abstract

The purpose of this study is to investigate physicians’ knowledge, attitudes and practice of generic medicine substitutions in China. We conducted a cross-sectional online questionnaire survey on physicians from secondary or tertiary hospitals in China from 2020 December to 2021 April. Descriptive statistical and ordered logistic regression were used for analysis. A total of 1225 physicians were included in the final analysis, and only 330 (26.94%) of them scored 4 or above in the knowledge part, which means that the physicians have a good knowledge of generic substitutions. Of the total, 586 (47.83%) agreed or strongly agreed that generic drugs could be substituted for originator drugs and 585 (47.75%) always or often prescribed generic medicines. The percentage of physicians with a positive attitude toward or that practice prescribing generic medicine is below 50%, which needs to be improved in China. Physicians’ knowledge, their attitude toward generic substitution, if familiar with the policy of generic substitution, and incentives for prescribing generic medicines are influencing factors for the practice of generic substitution. Our studies show that the practice of generic substitution by physicians could be improved by several measures in China. We suggested that the physicians should be taught more about the bulk-buy policy and the generic-originator equivalence evaluation policy. Moreover, government incentives to promote generic substitution should be established. Our study also suggested that physicians with less working experience and female physicians should learn more about generic substitution.

## 1. Introduction

To restrain the growing rate of health expenditure as well as improve access to treatment, generic substitution has been applied worldwide [1,2]. However, for the past few decades, the Chinese government had not required generic drugs to have the same quality and efficacy as the original drugs. In 2016, the Chinese government issued an announcement requiring quality and efficacy consistency assessment of generic drugs, as the first step to promoting generic substitution in China [3]. The volume-based purchase policy was then introduced in 2019 in China to reduce the price and to improve generic substitution. The mechanism of the policy allows the government to purchase medicines directly from pharmaceutical companies to reduce medicine prices through a strong government monopoly on demand. The government can ensure through regulations that generic medicine is distributed to hospitals. However, this kind of macro level policy may not achieve the goal of generic substitution in the Chinese market. Some policies to enhance the prescribing/dispensing of low-cost generics fail because the authorities have not taken into consideration all parties when developing the reforms, for example, in South Korea and Abu Dhabi [4,5,6]. To achieve successful generic substitution, the support of key stakeholders, including physicians, pharmacists and patients is pivotal [7]. Furthermore, a striking feature of the Chinese prescription drug market is that the physicians prescribe drugs by writing generic names but can choose a specific product (based on production company or drug price information) for patients in their prescription system. Thus, physicians play an important role in the generic substitution process in China. Moreover, policies have been implemented in China to improve medicine cost control and generic substitution, including the bulk-buy program and global budget. The bulk-buy program, in particular, could be combined with the consistency evaluation policy and promote generic substitution in China. Only medicines that pass the BE test can secure bulk-buy orders from the government. As mentioned, generic substitution is also affected by physicians. Macro level interventions and incentive policies that could encourage physicians to prescribe more generic medicines have not yet been introduced. Therefore, identifying the influencing factors of physicians’ prescription behavior toward generic medicines is important for further develop generic substitution intervention policy in China.

The knowledge, attitude and practice (KAP) is a model for changing human health-related behavior, and it is also a behavior intervention theory. It divides the change of human behavior into three continuous stages: acquiring knowledge, generating belief and forming behavior. Therefore, the knowledge of important stakeholders (especially the physicians) and attitudes towards generic substitution have been studied internationally [8,9,10,11]. Physicians’ KAP towards generic substitutions is an integral part of the successful enforcement of a generic prescribing policy [12,13,14]. It was noted that the opinion of physicians is a key influencing factor for patients’ views of generic medicine substitution [15,16]. Previous studies indicated that stakeholders, especially physicians had negative views of generics and resisted prescribing generic medicines. A systematic review of 52 studies from 11 countries found that 30% of the physicians regarded generics as lower quality, less effective and less safe than the originators [12]. However, to our knowledge, no peer-reviewed studies have assessed the KAP of healthcare professionals towards generic drugs at the beginning of the generic substitution policy in China. Thus, the aim of this study was to assess the level and factors associated with knowledge, attitudes and practice of generic substitution among physicians in China. We hypothesize that the effect of knowledge and attitudes on physician prescribing behavior is significant.

## 2. Methods

### 2.1. Participants

Participants were eligible to be included if they: (1) were working at secondary or tertiary hospitals during the study period (A secondary hospital is similar to a Regional hospital or District hospital in the West. A tertiary hospital is a comprehensive, referral, general hospital at the city, provincial or national level with a bed capacity exceeding 500); (2) speak the Chinese language; (3) have been in practice for at least 12 months in China. Pharmacists were excluded from the study. No incentives were provided to the enrolled participants.

### 2.2. Study Design

We conducted a cross-sectional online questionnaire survey from December 2020 to April 2021. Generally, a snowball sampling strategy was utilized to finalize the study. The WeChat app, which is the most widely used social networking app in China (over 1.1 billion users) was used for data collection. All authors of the present study initially posted the survey link or a quick response code of the questionnaire on their moments or social groups in WeChat. We describe this process in more detail below.

Due to the COVID-19 epidemic, we cannot conduct on-site investigations with doctors. Thus, our data consist of two parts. One part of our data is from 8 medical representatives (four from domestic companies; and four from foreign companies). The data were collected through WeChat. Medical representatives can usually contact physicians directly. Therefore, we compiled the survey on WeChat and then sent it to the physicians. They then sent the link to other physicians that they know to collect the data used in our study. They also encourage the physicians to send the survey to other physicians. Thus, we say that this is snowball sampling. The other part of the data is collected by a professional company (Wen Juan Xing. Changsha, China) that has a database of physicians. They perform a similar process to collect the survey data. The levels (secondary or tertiary) of the hospitals in which the physicians worked are also included in the physician database.

### 2.3. Survey Instrument

The questionnaire was adopted and customized from previous studies [17,18,19,20]. Two authors drafted the structured questionnaire, which was further reviewed and revised by two experts from the Shaanxi Provincial Center for Disease Control and Prevention for content validation. A pilot test (*n* = 30) was also performed to ensure the validity and interpretability of the questionnaire. The Cronbach alpha coefficient of our questionnaire in the pilot test was 0.79, indicating an acceptable level of internal consistency. The final version of the questionnaire was composed of two sections, including the demographics of the participants and their KAP toward generic substitution.

Demographic variables included age, gender, place of residence, province capital, education level, profession, work experience, hospital class and department. The KAP questionnaire contained three parts with 24 closed questions and two open questions. Part one investigated the respondents’ knowledge of generic medicines. This part was assessed through the respondents’ answers to the questions with ‘Yes’, ‘No’ or ‘Uncertain’. Each question has a correct answer. Each correct answer received 1 point and each mistake or unsure response received 0. There was a total of 6 questions. If a person answers all the questions correctly, they could achieve a score of 6. Part two assessing the attitude and part three assessing behavior were based on a 5-point Likert scale to evaluate attitudes and practice towards generic drug use, with answers ranging from ‘strongly agree’ to ‘strongly disagree’, or from “always” to “never”. In the attitude section, for the question ‘do you agree originator medicines can be substituted by generic medicines’, ‘strongly agree’ and ‘agree’ were taken as positive responses, ‘strongly disagree’ and ‘disagree’ were taken as negative responses, and ‘uncertain’ was regarded as neutral. In the behavior section, for the question ‘how often do you prescribe originator medicines’, ‘always’ and ‘often’ were both taken as positive responses, ‘never’ and ‘seldom’ were taken as negative responses and ‘sometimes’ was regarded as neutral. The results obtained from the pilot study were not included in the final results.

### 2.4. Data Analysis

Data were compiled in Microsoft Excel and analyzed by two co-investigators using STATA15.0. The results of the descriptive statistical analysis were reported as mean (SD) and median (interquartile range, IQR) values. Ordered logistic regression was conducted to determine the relationship between the variables. Four independents (knowledge, attitude and practice before and after 2018) are considered here. The independents of knowledge include demographic variables, the familiarity of policy, the attitude towards prescribing generic medicines to self/family and patients and the attitude toward incentive. The independents of attitude are similar to that of the knowledge in which the attitude is changed into knowledge of generic substitution. For practice, the independents include physicians’ socio-demographic characteristics, knowledge and attitude about generic substitution, familiarity with the policy and attitudes towards incentive. In all statistical analyses, the level of statistical significance was set at *p* < 0.05.

## 3. Results

The median time used by each participant filling in the questionnaire was 257 s (IQR, 181–381). In total, 1273 physicians participated in the study. Forty-eight questionnaires were subsequently excluded because of short response time, inconsistent logic and regular answers. Thus, the final sample size was 1225 in our study. Detailed demographic characteristics of the respondents are presented in Table 1. Among the participants, there were more females (52.90%) participants than male participants (47.10%). The majority of the participants (94.69%) were under 50 years old. More than half of them (53.22%) live in the capital of their province in China. A total of 93.55% of them had achieved a master’s degree (The statistics of the education level of physicians at the end of 2019: 39.2% bachelor degree or above, 39.1% junior college degree, 20.6% a technical secondary school degree, 1.1% a high school degree or below. The data were extracted from the “2019 Statistical Bulletin of Health Development” [21]). The positions of the participants are Primary (29.88%), Intermediate (44.00%) and Senior (26.12%). The statistical results for work experiences are ≤5 (28.33%), 6–10 (34.94%), 11–20 (25.55%), 21–30 (8.98%) and ≥31 (2.20%). Of the participants, 63.76% were from the tertiary hospital level (China has a hospital classification system in which a Tertiary level hospital is better than a Secondary level. Hospitals at the tertiary level are more multidisciplinary (please see Section 2.1 for more details)). A total of 446 (36.41%) were from eastern China, 356 (29.06%) from central, and 446 (36.41%) from western China. Physicians who worked in the internal medicine department made up 82.94% of the participants (The departments are divisions of the hospital according to function. They include internal medicine and surgery. Compared with the surgery department, the internal medicine department uses much more medicines. The department is defined as 1 if the physician is from internal medicine and the department is 0 if the physician is from surgery).

### 3.1. Knowledge

We designed the scores to evaluate the knowledge to range from 0 to 6. The mean score for the total sample was 2.84 (1.15). There are few differences between male and female responses. There were 330 (26.94%) respondents that achieved a score of 4 or above, demonstrating that less than one-third of respondents had a good knowledge of generic products. There were 59 (4.82%) respondents that achieved a score of 0. Moreover, no respondent achieved a score of 6 (Table 2). There were significant differences in knowledge scores between participants of different ages, working experience, professional title (*p* < 0.000), city of residence (*p* < 0.01), education level and hospital class (*p* < 0.05) (Table 1).

To figure out which factors are associated with the knowledge score, we performed an ordered logistic regression analysis. The results are as follows: If the physicians are in the capital (OR = 1.825; 95%CI: 1.065–3.129; *p* < 0.05), with more work experience (OR = 1.238; 95%CI: 1.008–1.520; *p* < 0.05) and familiar with the bulk buy policy (OR = 1.817; 95%CI: 1.387–2.379; *p* < 0.001) have significant positive relation with the knowledge score. OR is the odds ratio. Take the relationship between if the physicians are in the capital and knowledge for example. The OR is 1.825. This means that the physicians in the capital area will have a better knowledge about generic substitution. The OR number means that the percentage of good knowledge (Knowledge score is better than the mean score 2.84) among physicians in the capital area is EXP (1.825) = 6.2 times larger than that of the physicians outside the capital area. For work experience, the OR is 1.238. This means the percentage of good knowledge in the group of long working experience (the working experience is greater than the average working experience of 12 years) is EXP (1.238) = 3.4 times greater than that of the short working experience group. Other OR could also be explained by a similar process. If a physician’s attitude towards prescribing generic medicine to self/family members (OR = 1.227; 95%CI: 1.075–1.400; *p* < 0.05) is positive, the physician would have a good knowledge of generic substitution (Table 3).

### 3.2. Attitudes

Table 4 reported the frequency and percentage of responses for the statements about generic substitution attitudes. Physicians’ attitudes toward generic medicine substitution were determined by several factors, including the quality, therapeutic value, safety and price of the generic medicines. A large portion (812; 66.29%) of the responses agreed or strongly agreed that generic medicines have the same quality and therapeutic value as originator medicines and 21.31% of physicians had a neutral opinion about the differences in quality and therapeutic value between generic medicines and the originators. We found that 541 responses (accounting for 44.17%) agreed or strongly agreed that the safety of the generic medicines is similar to the originators. Concerning safety, 29.31% had a neutral attitude and the rest 26.53% had a negative attitude. We also studied the price difference between the originators and the generics. A total of 899 physicians (accounting for 73.39%) agreed or strongly disagreed that there is a price difference between the originators and the generics. Neutral opinions were held by 182 (14.86%) and the other 144 (13.88%) had negative opinions about the price difference between the originators and the generics.

The physician’s attitude towards generic substitution may be determined by whether they will prescribe generic medicines to self/family members. Thus, we have designed question 4 as shown in Table 4. More than half (725, 59.91%) agreed or strongly agreed that they prescribe the substitute generics to their own family members. Neutral opinions were held by 327 (26.69%) and the rest (173, 17.17%) had negative opinions. Profit incentive stimulation may be a very good policy to encourage physicians to prescribe generic medicines, and, thus, we designed question 5 as shown in Table 4. Totals of 525 (42.86%), 400 (32.65%) and 300 (24.49%) had positive, neutral and negative opinions, respectively. Finally, we designed a final attitude question considering if generic medicines can be substituted for the originators. Positive answers were given by 47.83% of the respondents, 29.31% of the respondents had a neutral attitude and 25.70% of the respondents disagreed or strongly disagreed that the generic medicines could be substituted for the originator medicines. Thus, in Table 2, we summarized the results of attitude. A total of 47.83% of physicians have a positive attitude while 22.86% of physicians have a negative attitude.

In order to figure out which factors are associated with attitude, as with knowledge, we performed ordered logistic regression analysis (Table 3). If the physicians live in the capital (OR = 1.763; 95%CI: 1.059–2.936; *p* < 0.05), their positive attitude toward prescribing generics to self and family numbers (OR = 3.280; 95%CI: 2.757–3.904; *p* < 0.001), the incentive stimulation (OR = 1.300; 95%CI: 1.145–1.475; *p* < 0.001) and familiar with the bulk buy policy (OR = 1.375; 95%CI: 1.087–1.739; *p* < 0.001) have significant positive relation with the attitude equivalent scores. However, female physicians show a negative attitude to generic substitution in China (OR = 0.799; 95%CI: 0.647–0.987; *p* < 0.05).

### 3.3. Practice

Practice describes the prescribing of generic medicines to patients. We divided the prescribing generic frequency into 5 levels. The scoring system 2, 1, 0, −1 and −2 represent always, often, sometimes, rarely and never, respectively. Moreover, because there are significant differences in quality between originators and generics since the generic substitution policy was established in 2018, we believe that there are also differences between practice before and after 2018. When assessing prescribing practice before 2018, nearly half of the participants 413 (23.71%) always or often prescribed generic medicines. Generic medicines were rarely or never prescribed by 285 (21.42%) of the participants and 427 (34.86%) of the physicians prescribed the generics occasionally. When assessing prescribing practice after 2018, nearly half of the participants, 585 (47.75%), always or often prescribed generic medicines. Of the participants, 283 (23.10%) rarely or never prescribe generic medicines and 357 (29.14%) participants prescribe the generics occasionally. Compared with the practice pattern before 2018, there is a positive change in the practice of prescribing generics after 2018. Detailed frequencies of responses about practices are described in Table 2.

Accordingly, our analysis indicated that gender, age, knowledge about generics, the attitude about generic substitution, profit incentive and bulk buy policy are the independent influencing factors for generic prescription frequency (Table 5). Female physicians were found to prescribe less generic medicines than males (OR = 0.728; 95%CI: 0.586–0.904; *p* < 0.001). Physicians’ age was also positively associated with generic prescription frequency (OR = 1.276; 95%CI: 1.006–1.620; *p* < 0.05) after 2018. Furthermore, if a physician has a good knowledge and attitude about generic substitution, he or she will prescribe generic medicines (OR = 1.119; 95%CI: 1.015–1.233; *p* < 0.05. OR = 1.360; 95%CI: 1.179–1.569, *p* < 0.001). The incentive stimulation has a significant positive effect on the physician prescribing generic medicines (OR = 1.308; 95%CI: 1.157–1.480; *p* < 0.001). If a physician knows more about the generic substitution policy established by the government, he or she would be more interested in prescribing generic medicines (OR = 3.544; 95%CI: 2.780–4.518; *p* < 0.001). Note that the comparison before 2018 and after 2018 is a robustness test of our result, most of the factors significant in model 4 were identified in practice before 2018 (model 3).

The open questions mainly focus on the issue of generic substitution and volume-based policy from the perspective of physicians. Our results show that quality and safety is still the main problem considered by the physicians in China. The second issue of generic substitution is the knowledge gap between the physicians and the patients. Policy uncertainty is the main issue facing the policy.

## 4. Discussion

Our study findings suggested that 71.47% of the participants achieved a knowledge score equal to or less than 3, which means that most of the participants have poor knowledge of generic medicines. With the poor knowledge of generic substitution among the physicians, only 45.41% of the participants agreed or strongly agreed that generic medicines could be substituted for the originators. Only 46.20% of the participants frequently or often prescribe generic medicines. Our findings suggest that knowledge, attitude, profit incentive and government policy were the factors associated with physicians’ prescribing behavior in China. Furthermore, the main factors impacting physicians’ knowledge and attitude about generics were the government policy and the willingness to switch generic medicines to themselves or their family members.

Although physicians’ knowledge, attitudes and practice of generic drug substitution generally varied by country due to differences in culture and policies, good knowledge of and attitude towards generic drugs could boost generic prescription practice. For comparision, we have studied the KAP of generic substitution in several countries.

In terms of knowledge of generic drugs, physicians from countries with high income have better generic substitution knowledge than physicians from countries with low income. It was reported that physicians from Saudi Arabia and Slovenia mostly claimed to have enough knowledge of generic substitution [17,18]. The findings of this study were in line with those of other published studies on low-income countries. A cross-sectional survey study in India focused on doctors from tertiary-care teaching hospitals, showed that participants had poor knowledge of generic drugs [19]. Many of them had misunderstandings about the similarity of active ingredients, therapeutic equivalence and quality consistency evaluation of generic medicines [19]. Two studies from Pakistan and Nigeria also generated similar results, demonstrating that the practitioners had incorrect knowledge of the therapeutic effect of generic medicines [20,22]. Chinese physicians have poor knowledge because the generic–originator equivalence evaluation policy was only established in 2018.

The doubts of physicians regarding the quality and safety of generic substitutes in the present study are consistent with those revealed by other studies conducted on general physicians [23,24,25]. Colgan et al. collated international studies published after 1980 and integrated the collective views of physicians in a comprehensive narrative review and concluded that a significant proportion of doctors hold negative perceptions of generic medicines [12]. A meaningful proportion of physicians expressed negative perceptions about generic medicines in the United States, the country where the earliest generic substitution policy was implemented, representing a potential barrier to generic use [26]. Furthermore, physicians were more cautious when using antiepileptic and anticoagulation generics [27,28]. However, considering physicians’ attitudes in China, nearly half of the respondents agree with generic substitution. This is mainly because the volume-based policy was introduced in China recently. In India, physicians generally showed a positive attitude toward the safety, efficacy and quality of generic medicines [11]. Another study using one-to-one semi-structured interviews found that general practitioners in Ireland had positive attitudes towards generic medicines [29]. Similarly, the majority of doctors in both Greece and Cyprus agreed that the effectiveness, safety and efficacy of generic drugs may not be excellent but it was acceptable [30].

The US government’s policy of mandatory generic drug prescription had a positive effect on the US generic substitution process. In France, Beatrice Rine studied physicians’ practice of generic substitution in the district with the lowest generic substitution. Results show that most of the physicians are not against the generic substitution. However, due to the lack of evidence for the equivalence between generics and originators, the physicians find it difficult to persuade patients who have a negative attitude towards generic medicines [31]. These challenges produce negative effects on the prescribing behavior of physicians in France [31].

Furthermore, our results indicated that physicians in China preferred prescribing originator medicines. This is because physicians believe that originator medicines are of better quality and more effective in treatment. On the other hand, the price of originator medicine is higher and prescribing such medicines produces greater economic benefits for the hospital and the prescriber [31]. Thus, the physicians did not usually prescribe generics in their clinical practice. Our findings suggest that the factors most associated with the practice of prescribing generic substitutions are knowledge, attitude, incentive stimulation and attitude towards prescribing generic substitution to self/family.

Considering these points, to encourage both physicians and patients to accept generic substitutions they should be educated about the improvements in the quality of generic medicines. Policies should be established by the government to stimulate the substitution. In China, due to the history of low-quality generic medicine, physicians have a very low willingness for generic substitution. We believe that the generic substitution status can be improved through the originator and generic equivalence evaluation.

France, Ireland, Portugal and Sweden introduced education as one main of demand-side measures to improve physicians’ knowledge and attitudes toward generic substitution [32]. Thus, more individual incentive measures could encourage physicians to prescribe generic medicines through the macro policy that was introduced in China. First, an education program should be introduced in China to encourage physicians to prescribe generics in their practice. Second, individual incentives for physicians and pharmacists should also be considered. Some research has shown that policies to enhance the prescribing/dispensing of low-cost generics fail because the authorities have not taken into consideration all parties when developing the reforms [4,5,6,33]. Finally, other factors that impact the prescribing behavior of the physician should be researched in China.

This study has some limitations. Our questionnaire did not include all of the factors affecting general physician prescribing behavior. However, as a baseline study, it will provide a foundation to further explore physician behavior. In addition, our study was a cross-sectional survey representing one point in time, and may not reflect any dynamic changes in respondent awareness of the use of generics. Finally, a further study of generic substitutions for certain medications and their alternatives, e.g., blood pressure or cardiovascular disease, would be very significant because many people suffer from these diseases and they are a heavy burden for society. The study of generic substitution in treating these kinds of diseases could lower the burden for society.

## 5. Conclusions

This paper first measured the physician’s situation and KAP factors of prescription decision considering the choice between generic and brand-name drugs in China. In conclusion, we found that Chinese physicians do not have a good knowledge of generic substitution and the attitudes toward and practice of generic substitution are poor. We believe that this is because the generic substitution policies just began recently. This study indicates that the important factors are a national policy for generic substitution, physicians’ knowledge, physicians’ attitudes and incentives for physicians to prescribe generic medicines. We suggested that physicians’ knowledge and attitude toward generic substitution could be improved through educational programs and government incentive policies should be established to promote generic substitution in China.

## Figures and Tables

**Table 1 ijerph-18-07749-t001:** Demographic characteristics of physicians and their knowledge scores.

Characteristics	N (%)	Mean Score (SD)	Median Score (IQR)	*χ* ^2^	*p*-Value
Total		2.84 (1.15)	3 (2,4)		
Gender				7.38	0.194
Male	577 (47.10)	2.84 (1.20)	3 (2,4)		
Female	648 (52.90)	2.83 (1.11)	3 (2,4)		
Age (years)				49.35	0.000
18–30	342 (27.92)	2.59 (1.17)	3 (2,3)		
31–40	582 (47.51)	2.84 (1.15)	3 (2,4)		
41–50	236 (19.27)	3.13 (1.07)	3 (3,4)		
51–60	55 (4.49)	3.00 (1.11)	3 (2,4)		
≥61	10 (0.82)	3.30 (1.42)	4 (3,4)		
Place of residence				21.62	0.001
Capital	652 (53.22)	2.95 (1.09)	3 (2,4)		
Other	573 (46.78)	2.71 (1.26)	3 (2,3)		
Education level				28.96	0.016
College or below	48 (3.93)	2.63 (1.41)	3 (2,3.5)		
Bachelor	694 (56.65)	2.73 (1.17)	3 (2,3)		
Master	404 (32.98)	3.00 (1.08)	3 (2.5,4)		
Ph. D	79 (6.45)	3.08 (1.11)	3 (2,4)		
Position				52.42	0.000
Primary	366 (29.88)	2.56 (1.18)	3 (2,3)		
Intermediate	539 (44.00)	2.84 (1.15)	3 (2,4)		
Senior	320 (26.12)	3.15 (1.05)	3 (3,4)		
Work experience (years)				48.92	0.000
≤5	347 (28.33)	2.62 (1.19)	3 (2,3)		
6–10	428 (34.94)	2.79 (1.16)	3 (2,3)		
11–20	313 (25.55)	2.96 (1.10)	3 (2,4)		
21–30	110 (8.98)	3.20 (1.05)	3 (3,4)		
≥31	27 (2.20)	3.30 (1.03)	4 (3,4)		
Hospital class				22.02	0.001
Secondary	444 (36.21)	2.65 (1.18)	3 (2,3)		
Tertiary	781 (63.76)	2.94 (1.13)	3 (2,4)		
Department				4.455	0.486
Internal medicine	1016 (82.94)	2.82 (1.16)	3 (2,4)		
Other	209 (17.06)	2.93 (1.14)	3 (2,4)		
Location				13.266	0.209
East	423 (34.53)	2.79 (1.15)	3 (2,3)		
Middle	356 (29.06)	2.93 (1.07)	3 (2,4)		
West	446 (36.41)	2.81 (1.22)	3 (2,4)		

SD: standard deviation; IQR: interquartile range.

**Table 2 ijerph-18-07749-t002:** Physician generic substitution KAP distribution in China (*n* = 1225).

KAP	N/%	Cum.	KAP	N/%	Cum.
**Knowledge score**			**Practice (before 2018)**		
0	59 (4.82)	4.82	Never	101 (8.24)	8.24
1	92 (7.51)	12.33	Rarely	284 (23.18)	31.43
2	236 (19.27)	31.59	Sometime	427 (34.86)	66.29
3	508 (41.47)	73.06	Often	350 (28.57)	94.86
4	264 (21.55)	94.61	Always	63 (5.14)	100.00
5	66 (5.39)	100.00	**Practice (after 2018)**		
6	0 (0.00)	100.00	Never	82 (6.69)	6.69
**Attitude**			Rarely	201 (16.41)	23.10
Positive	586 (47.83)	47.83	Sometime	357 (29.14)	52.24
Neutral	359 (29.31)	78.14	Often	493 (40.24)	92.49
Negative	280 (22.86)	100.00	Always	92 (7.51)	100.00

KAP: Knowledge-Attitude-Practice.

**Table 3 ijerph-18-07749-t003:** Ordered logistic regression analysis of factors associated with knowledge score and attitude.

Variables	Model 1	Model 2
	Knowledge Score	Attitude
Gender	1.067	0.799 **
	(0.864–1.316)	(0.647–0.987)
Age	1.036	1.207
	(0.822–1.305)	(0.966–1.508)
Education	1.116	0.998
	(0.925–1.346)	(0.837–1.190)
Work experience	1.238 **	0.906
	(1.008–1.520)	(0.757–1.085)
Department	0.861	1.239
	(0.649–1.144)	(0.911–1.685)
Position	1.149	1.230
	(0.912–1.448)	(0.993–1.523)
Hospital class	1.211	1.016
	(0.957–1.532)	(0.801–1.288)
Capital	1.825 **	1.763 **
	(1.065–3.129)	(1.059–2.936)
Income	0.935	0.766 **
	(0.732–1.193)	(0.612–0.958)
Attitude	1.116	NA
	(0.987–1.261)	
Knowledge	NA	1.088
		(0.994–1.192)
Attitude to self/family member	1.227 **	3.280 ***
	(1.075–1.400)	(2.757–3.904)
Attitude to profit incentive	1.002	1.300 ***
	(0.899–1.117)	(1.145–1.475)
Familiarity of the Policy	1.817 ***	1.375 ***
	(1.387–2.379)	(1.087–1.739)
East	0.841	1.123
	(0.654–1.082)	(0.876–1.438)
Middle	1.008	1.204
	(0.771–1.318)	(0.920–1.576)
Cutpoint 1	−0.799	−2.062
	(−1.555–−0.0427)	(−2.880–−1.244)
Cutpoint 2	0.236	0.390
	(−0.491–0.963)	(−0.383–1.162)
Cutpoint 3	1.459	2.197
	(0.736–2.182)	(1.415–2.978)
Cutpoint 4	3.344	4.607
	(2.595–4.092)	(3.786–5.429)
Cutpoint 5	5.314	NA
	(4.520–6.107)	
Pseudo-R^2^	0.040	
Observations	1225	1225

Note: OR, odds ratio (we have explained the OR in the first paragraph following Table 1.); CI, confidence interval; Goodness of fit of the model: *χ*^2^ = 133.67, *p* < 0.001, *χ*^2^ = 330.49, *p* < 0.001 *** *p* < 0.01, ** *p* < 0.05.

**Table 4 ijerph-18-07749-t004:** Physician’s attitudes of statement on generic substitution.

	N (%)				
	Strongly Agree	Agree	Neutral	Disagree	Strongly Disagree
(1) The generic drugs have the same quality and efficacy as originator drugs.	247 (20.16)	565 (46.12)	261 (21.31)	132 (10.78)	20 (1.63)
(2) There is a difference between the safety of the generics and the originators.	126 (10.29)	415 (33.88)	359 (29.31)	290 (23.67)	35 (2.86)
(3) There is a certain price difference between the originators and the generics.	311 (25.39)	588 (48.00)	182 (14.86)	118 (9.63)	26 (2.12)
(4) Could you prescribe the substitute generics for your own or family member?	139 (11.35)	586 (47.84)	327 (26.69)	129 (10.53)	44 (3.59)
(5) The hospital should give the doctors profit incentives for prescribing the generics.	126 (10.29)	399 (32.57)	400 (32.65)	245 (20.00)	55 (4.49)
(6) Do you think that the originators can be substituted by the generics?	131 (10.69)	455 (37.14)	359 (29.31)	233 (19.02)	47 (3.84)

**Table 5 ijerph-18-07749-t005:** Ordered logistic regression analysis of factors associated with physicians prescribing generics.

Variables	Model 3	Model 4
	Practice before 2018	Practice after 2018
Gender	0.638 ***	0.728 ***
	(0.516–0.788)	(0.586–0.904)
Age	1.156	1.276 **
	(0.923–1.448)	(1.006–1.620)
Education	0.915	1.062
	(0.764–1.095)	(0.884–1.274)
Work experience	1.055	0.992
	(0.873–1.276)	(0.810–1.214)
Department	1.367 **	1.229
	(1.038–1.799)	(0.919–1.643)
Position	1.084	0.942
	(0.863–1.363)	(0.741–1.197)
Hospital class	0.848	0.823
	(0.672–1.069)	(0.647–1.047)
Capital	1.302	0.919
	(0.759–2.233)	(0.529–1.598)
Income	0.944	1.092
	(0.740–1.203)	(0.847–1.407)
Knowledge	1.034	1.119 **
	(0.935–1.144)	(1.015–1.233)
Attitude	1.329 ***	1.360 ***
	(1.159–1.525)	(1.179–1.569)
Attitude to self/family member	1.473 ***	1.575 ***
	(1.276– 1.700)	(1.349–1.839)
Attitude to profit incentive	1.323 ***	1.308 ***
	(1.169–1.499)	(1.157–1.480)
Familiarity of the Policy	NA	3.544 ***
		(2.780–4.518)
East	0.763 **	0.840
	(0.587–0.990)	(0.651–1.083)
Middle	0.934	0.921
	(0.724–1.206)	(0.700–1.210)
Cutpoint 1	−2.976	−1.258
	(−3.990–−1.963)	(−2.042–−0.474)
Cutpoint 2	−0.844	0.548
	(−1.809–0.121)	(−0.214–1.310)
Cutpoint 3	0.823	2.210
	(−0.141–1.787)	(1.440–2.979)
Cutpoint 4	3.263	5.007
	(2.271–4.254)	(4.182–5.832)
Pseudo-R2	0.046	0.109
Observations	1225	1225

Note: OR, odds ratio; CI, confidence interval; Goodness of fit of the model: *χ*^2^ = 184.60, *p* < 0.001; *χ*^2^ = 348.42, *p* < 0.001; *** *p* < 0.01, ** *p* < 0.05.

## Data Availability

The data analyzed in this study were obtained from online-servey. This data can be found here: Requests to access these datasets should be directed to Yu Fang at yufang@mail.xjtu.edu.cn.

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
