# Peer review of "Physicians’ Knowledge, Attitude and Practice of Generic Substitution in China: A Cross-Sectional Online Survey"

_ijerph, 2021, doi:10.3390/ijerph18157749_

Round 1
Reviewer 1 Report
Please see the attached PDF file.

Author Response
Main Comments:
- The authors used a snowball sampling strategy via a social media app to solicit participants. Given that the target population is clear (the physicians who speak Chinese and work at a secondary or a tertiary hospital for at least a year), I am not sure the snowball sampling via a social media app is appropriate. Who responded to the survey? How did those who responded recruit the other participants (i.e., the snowball effect)? I am concerned with the resulting sample because it is hard to believe that a person without a post-baccalaureate degree can be a physician in China. See Table 2, where it shows 694 + 48 respondents with bachelor’s degrees or no college degrees. The reviewer also concerns about the degree problem. Since China is a developing country, a large portion of the physicians who are very old don’t own a degree higher than bachelor. That is because we only enlarged the group of degree beyond bachelor since 2000. Thus, we show that a large part of the physicians don’t have a higher degree than bachelor. Even parts of the physicians own a degree lower than bachelor. We have wright more in detail in our manuscript.? • “This part was assessed using a three-point scale: ‘Yes’, ‘No’ or ‘Uncertain’ (page 2, last line). ? • “We designed the scores to evaluate the knowledges to range from 0 to 6.” (Section 3.1, Line 1) Part one investigated the respondents’ knowledge about generics medicines. This part was assessed through the respondents’ answers to the questions with ‘Yes’, ‘No’ or ‘Uncertain’. Each question has a correct answer. 1 point was given for each correct answer and 0 for each mistake or unsure response. The total questions are 6. If a person answers all the questions correctly, he or she could get a score of 6. Part two for the attitude and three for the behavior sections were based on a 5-point Likert scale to evaluate attitudes and practice towards generic drug use, with answers ranging from ‘strongly agree’ to ‘strongly disagree’, or from “always” to “never”. In the attitude section, for question ‘do you agree originator medicines can be substituted by generic medicines’, ‘strongly agree’ and ‘agree’ were taken as positive response, ‘strongly disagree’ and ‘disagree’ were taken as negative response, and “uncertain’ was regarded as neutrality. In the behavior section, for question ‘how often do you prescribe the originator medicines’, ‘always’ and ‘often’ were both taken as positive response, ‘never’ and ‘seldom’ were supposed as negative response, and ‘sometimes’ was regarded as neutrality. The results obtained from pilot study were not included in the final results.
- Response: Thanks for your comments! We have made the process more clearly as follows:
- It is confusing how the authors went from ‘Yes/No’ to “0 to 6” and why. It may make sense if the questionnaire has been included.
- ? • “1 point was given for each correct answer and 0 for each mistake or unsure response” (Section 2.4, Line 2)
- 2. The data collected and described in Sections 2 and 3 can be rephrased for clearer understanding. Take the “knowledge” variable as an example:
- Response:Thanks for your comment! Due to the COVID-19 epidemic, we cannot conduct on-site investigations with doctors. Thus, our data is from two part. One part of our data is from 8 medical representatives (four from domestic company; and four from the foreign company). The data are collected through WeChat which is a famous social media app in China. Usually the medical representatives can connect with physicians directly. So we made a survey on WeChat and then we send them to the physicians. They will finally send the link to physicians they know to collect the data used in our manuscript. They also encourage the physicians to send the survey to other physicians. Thus, we say that this is the snowball sampling. The other part of the data is collected by a professional company (Wen Juan Xing) which owns a list of physicians in their data base. They do a similar process to collect the survey data. The levels(secondary or tertiary) of the hospitals in which the physicians worked are also included in the physician’s data base.
- The interpretation of the results from the ordered logistic regression can be improved. The estimated coefficient is an odds ratio, not the difference in quantity. For example, “female physicians were found to prescribing less generic medicines than males (OR = 0.728; 95%CI: 0.586-0.904; p<0.001)” (page 8, 1st paragraph, lines 5-6). The value 0.728 is an odds ratio, and its significance implies that a female physician is less likely to prescribe generic medicines than her male counterpart. That is, a male physician is 2 times more likely to prescribe than his female counterpart ( i.e., Pr( male prescribes ) / Pr( female prescribes ) = EXP(0.728) = 2.07.).
- Response: Thanks for your comments! We agree that we need to explain more about the odds ratio. We have edited the odd ratios in our manuscript to give more explain about the odd ratios based on your advice. Please see the paragraph under table 1.
- Finally, I recommend the study to focus on a few generic substitutions, e.g., blood pressure medications and their alternatives, in order to increase the validity of the study. The discussion on physicians’ attitudes and practices towards the generic ones would shed more light when conditioning on the same generic substitutions. Minor Comments: Response: Thanks for your comments! We have made this concept more clearly in our manuscript. Please see section 2.1.2. Please proofread the manuscript. For example (this is just a shortlist of examples): • “nagetive” (Section 2.4, End of Line 5) ? • Extra space between “level” and “and” (page 4, 4th line) Response: Thanks for your comments! We have done a proofreading of our manuscript and we have corrected the mistakes.
- ? • “don’t owns a good” (1st line in Section 5)
- ? • “substation” (3rd line from the bottom in Section 2.4)
- • “ordered logistic regression”, not “order”
- A secondary hospital is similar to a Regional hospital or District hospital. A tertiary hospital is a comprehensive, referral, general hospitals at the city, provincial or national level with a bed capacity exceeding 500.
- 1. How are secondary and tertiary hospitals defined in the Chinese healthcare system?
- Response: Thanks for your comments! We think that your advices are very good! We have included your advices in the last paragraph of section “Discussion”. This is a limitation of our study. Further study of the generic substitution for a variety of medications may be the next study topic for us.
- End of the paragraph below Table 1 on page 4. The discussion in this paragraph has nothing to do with Table 3.
- Response: Thanks for your comments! We have done a proofreading of our manuscript and we have corrected the similar mistakes.
- Table 2. What is the “Statistics” shown? The chi-square statistic?
- Response: Thanks for your comments! The statistics shown is chi-square statistic. Please see table 2.
- Tables 4 and 5. Please provide pseudo-R2 and cutpoints.
- Response: Thanks for your comments! We have provided pseudo R2 and cutpoints in table 4 and 5.
- Footnotes of Tables 4 and 5. “Robust standard errors in parentheses.” Aren’t the values in parentheses the lower and the upper bounds of 95% CI?
- Response: Thanks for your comments! We have discard the robust standard errors. Please see table 4 and 5.
- Are there any interesting findings from the two open questions?
Response: Thanks for your comments! The open questions are mainly focus on the issue of generic substitution and volume based policy from the perspective of physicians. Our result show that the quality and safety still is the main problem considered from the physicians in China. The second issue of generic substitution is that the knowledge gap be-tween the physicians and the patients. The main issue of the policy is that the policy uncertainty.
We have included it at the end of the results section.
Reviewer 2 Report
Dear Authors
I carefully evaluated the present paper, finding it overall interesting, and well written. The theme is interesting, but the manuscript needs some improvements.
The introduction is overall well written, but the theoretical background needs some improvements. Study hypothesis and study motivation should be presented more in details. Moreover, I suggest to focus not only on the local Chinese context, but move to a global context. In fact, medicine substitution and brand issues are arguments of main interest worldwide. As a consequence, you should start from the local context to reach a general context, especially where the same problem occurs whit medical substitution. As a consequences, you should frame the problem in a global context, also giving some example in other countries.
Discussion should be changed accordingly, discussing the generalisability (external validity) of the study results. More in general, all the findings have to be compared with the field literature, also providing a general explanation and a motivation for those findings.
The generalizability of your results should be improved. Are these finding relevant also out of a local context?
In the conclusion section, state clearly what this paper adds to knowledge about the theme, with respect to previous published articles. The novelty of your findings remains questionable. How your results contribute to bridge a literature gap?
Best regards
Round 2
Reviewer 1 Report
Comments on “The Physicians’ Knowledge, Attitude and Practice of Generic Substitution in China: a Cross-sectional Online Survey” (IJERPH# 1233615-revised)
1.I included the authors’ responseson pages 2 –3.The responses were presented in a rather confusing format and I am not sure if there is any system error when submitting the responses.It is really difficult to read through the authors’ responses.
2.Please proofread. Here are just a few examples:
•(Line 2, page 3): “doc-tors”•(Line 2, page 3): “two parts”
•(Line 3, page 3): “domestic companies”and “foreign companies”
•(Table 1, footnote 2) “statistics”
3.Please provide the reference to “the health statistic year book”(Table 1, footnote 2). The reader might be interested in getting relevant data from this official site.I know of National Bureau of Statistics of China, China Statistical Yearbook, China Health and Family Planning Statistical Yearbook. But I could not find “the health statistic year book.”
Reviewer 2 Report
Dear Authors
The paper has been substantially improved with respect to the first version.
Best regards
Author Response
Thank for your comments to improve the utility of the manuscript.